Use of video surveillance to measure the influences of habitat management and landscape composition on pollinator visitation and pollen deposition in pumpkin (Cucurbita pepo) agroecosystems

Phillips Benjamin W. 1
Gardiner Mary M. gardiner.29@osu.edu
Department of Entomology, The Ohio State University—Ohio Agricultural Research and Development Center , Wooster, OH , United States
1 Current affiliation: Michigan State University , Saginaw, MI , United States
Jonsson Mattias
Electronic publication date: 2015 Nov 5
Publication date: 2015
Volume: 3
Electronic Location ID: e1342
Received 2015 Jul 22; Accepted 2015 Sep 30
Copyright: © 2015 Phillips and Gardiner
Copyright year: 2015
Copyright holder: Phillips and Gardiner
License: This is an open access article distributed under the terms of the Creative Commons Attribution License, which permits unrestricted use, distribution, reproduction and adaptation in any medium and for any purpose provided that it is properly attributed. For attribution, the original author(s), title, publication source (PeerJ) and either DOI or URL of the article must be cited.
License URL: https://creativecommons.org/licenses/by/4.0/

Keywords: Pollination services, Landscape, Pumpkin, Peponapis pruinosa, Bumble bee, Apis mellifera, Habitat managment, Floral strips

Funding: USDA NRCS Conservation Innovation Grant 69-A75-9-204 NCR-SARE Graduate Student Grant Funding was provided by a USDA NRCS Conservation Innovation Grant Program to the Pollinator Partnership and MMG (69-A75-9-204) and a NCR-SARE Graduate Student Grant to BWP. The funders had no role in study design, data collection and analysis, decision to publish, or preparation of the manuscript.

==============================
Pumpkin (Cucurbita pepo) production relies on insect-mediated pollination, which is provided by managed and wild pollinators. The goals of this study were to measure the visitation frequency, longevity and temporal activity patterns of pumpkin pollinators and to determine if local habitat management and landscape composition affected this pollination service. We used video surveillance to monitor bee acitivty within male and female pumpkin flowers in 2011 and 2012 across a pollination window of 0600–1200 h. We also quantified the amount of pollen deposited in female flowers across this time period. In 2011, A. mellifera made significantly more floral visits than other bees, and in 2012 Bombus spp. was the dominant pumpkin pollinator. We found variation in visitation among male and female pumpkin flowers, with A. mellifera visiting female flowers more often and spending longer per visit within them than male flowers in both 2011 and 2012. The squash bee P. pruinosa visited male flowers more frequently in 2012, but individuals spent equal time in both flower sexes. We did not find variation in the timing of flower visitation among species across the observed pollination window. In both 2011 and 2012 we found that the majority of pollen deposition occurred within the first two hours (0600–0800 h) of observation; there was no difference between the pollen deposited during this two-hour period and full pollination window (0600–1200 h). Local additions of sweet alyssum floral strips or a field buffer strip of native wildflowers did not have an effect on the foraging activity of bees or pollen deposition. However, semi-natural and urban habitats in the surrounding landscape were positively correlated with the frequency of flower visitation by wild pollinators and the amount of pollen deposited within female flowers.

Introduction

Worldwide, 35% of the global food supply is highly reliant on animal-mediated pollination services (Klein et al., 2007; Nicholls & Altieri, 2013). In the United States alone, pollinators account for 40 billion USD per year in fruit, fiber, vegetable and legume crops (Pimentel et al., 1997), with an estimated 1.6–14.8 billion USD of that attributed to the honey bee, Apis mellifera L. (Hymenoptera: Apidae) (Southwick & Southwick, 1992; Morse & Calderone, 2000; Losey & Vaughan, 2006). Across the United States and Europe, severe declines in the supply of honey bees for crop pollination have occurred as a result of colony collapse disorder (Allen-Wardell et al., 1998; Aizen & Harder, 2009; Potts et al., 2010a; Potts et al., 2010b). Wild bee species also contribute significantly to pollination within many cropping systems (Stanghellini, Ambrose & Schultheis, 1998; Kremen, Williams & Thorp, 2002; Winfree et al., 2007; Garibaldi et al., 2013; Garibaldi et al., 2014). Unfortunately, several wild pollinator taxa—such as some bumble bee species—have also exhibited significant declines in richness and abundance, further threatening the continued supply of pollination services to agroecosystems (Goulson, Lye & Darvill, 2008; Cameron et al., 2011).

Habitat management to support pollinators

Several potential drivers of population decline among pollinators have been identified, including pesticide use (Sanchez-Bayo & Goka, 2014; Rundlof et al., 2015), pathogen and parasite infection (Meeus et al., 2011; Blaker et al., 2014; Fuerst et al., 2014; Goulson et al., 2015; McMahon et al., 2015), exposure to heavy metals (Moran et al., 2012), climate change, land use change and fragmentation of pollinator habitat, or a combination of several factors (Potts et al., 2010a; Gonzalez-Varo et al., 2013; Rader et al., 2013; Vanbergen et al., 2013; Rands, 2014; Scheper et al., 2014; Goulson et al., 2015; Rollin et al., 2015). To address the impacts agricultural intensification may have on wild and managed bee populations, agri-environmental schemes have been designed to reestablish pollinator resources within agricultural landscapes (Haaland, Naisbit & Bersier, 2011; Rollin et al., 2015). Enhancing farmscape-scale heterogeneity through this form of habitat management has been demonstrated to increase pollinator richness by providing resources across time and space (Klein, 2011; Kennedy et al., 2013; Shackelford et al., 2013; Blaauw & Isaacs, 2014; Garibaldi et al., 2014). The flowering plants established in these plantings have been shown to be highly attractive to a diversity of beneficial insects, increasing fecundity, longevity, and the ecosystem services provided such as pollination and biological control (Baggen & Gurr, 1998; Johanowicz & Mitchell, 2000; Landis, Wratten & Gurr, 2000; Pontin et al., 2005; Lee, Andow & Heimpel, 2006; Pywell et al., 2006; Tuell et al., 2008). However, the addition of floral resources could in theory result in competition for pollinators with the target crop (Zhang et al., 2007). Thus, evaluation of the impacts of these strategies on foraging efficiency within specific agroecosystems is a necessary step towards incorporation of this conservation practice.

Landscape context can influence the outcome of habitat management

When habitat management practices are incorporated into a farmscape, larger scale landscape composition and heterogeneity can influence the pool of beneficial species supplied to an established planting and the arthropod mediated ecosystem services they are able to support in nearby farm fields (Isaacs et al., 2009; Batáry et al., 2011; Concepción et al., 2012; Rodriguez-Saona, Blaauw & Isaacs, 2012; Tscharntke et al., 2012). In synthesis papers, Ricketts et al. (2008) and Garibaldi et al. (2011) found decreased stability and levels of pollination services provided by pollinator communities with increasing distance from natural areas. Kennedy et al. (2013) analyzed data from 39 studies focusing on 23 cropping systems and found that organically-managed cropping systems supported a greater abundance and richness of wild bees. Similar to previous reviews, they also documented that at landscape scales the proportion of high-quality natural habitat was positively related to bee abundance and richness. Further, landscape factors have been shown to mediate the impact of some agricultural inputs. For example, Park et al. (2015) found that pesticide impacts on wild bees in apple orchards were reduced in landscapes with high proprotions of natural habitat.

Habitat management in cucurbit agroecosystems

As agricultural intensification threatens both natural pest control and pollination, habitat management strategies often target multiple key insect guilds (Campbell et al., 2012). The sustainability of pumpkin, Cucurbita pepo L. (Cucurbitales, Cucurbitaceae), production relies in part on biological control to suppress key pests. Being a monoecious crop, pumpkin is also dependent on insect-mediated pollination (Wien, 1997). Furthermore, pumpkin provides a unique study system to evaluate habitat management in sustaining pollination services because they are visited by managed (A. mellifera) and wild (Bombus spp.) social bees as well as a wild solitary specialist pollinator, Peponapis pruinosa (Say) (Hymenoptera: Apidae) (Hurd, Linsley & Whitaker, 1971; Hurd, Linsley & Michelbacher, 1974). Due to differences in foraging traits, greater pollinator richness within this system may lead to functional complementarity or synergy, thereby improving pollination efficiency (Bluthgen & Klein, 2011).

In a network of pumpkin farms across central and southern Ohio, we completed the following research objectives: (1) Use video surveillance to measure the relative contribution of pumpkin pollinator taxa to pollination services; (2) Determine if pollinators varied in their visitation frequency and visit longevity in male and female flowers; (3) Examine if temporal complementarity exists among flower visitation by pumpkin pollinators; and (4) Determine how habitat management and landscape composition influence pollination visitation and pollen deposition. This study was completed during the growing seasons of 2011 and 2012. In 2011, we measured how landscape composition influenced pollinator activity and pollination services within pumpkin crops. In 2012, we added habitat management as a variable and evaluated how the addition of floral strips of sweet alyssum Lobularia maritima (L.) (Brassicales: Brassicaceae) or a buffer strip of native perennial wildflowers and grasses—as well as the surrounding landscape—influenced pollinators and their function.

Methods

Study sites

Ohio is the 2nd largest pumpkin-producing state in the United States, and two regions within Ohio were selected that represent major production areas (USDA-NASS, 2013). In 2011, 12 farms were included in our study; six in Wayne, Stark, and Medina counties in northern Ohio, and six in Jackson, Pike, Highland, and Warren counties in southern Ohio (Table 1, Fig. 1A). In 2012, 15 farms were included, with eight farms in northern Ohio and seven in southern Ohio (Table 1, Fig. 1B). The distance between the two closest farms was 4.25 km within a given year. Farms were chosen based on grower interest in participating and by assessment of the composition of habitats in the surrounding landscape. One to four Apis mellifera hives were located within each farm.

Figure 1 Pumpkin sites were located in growing regions in northern and southern Ohio.

In 2011, we established 12 pumpkins sites on individual farms. We did not evaluate habitat management in 2011; each pumpkin site was adjacent to a grassy field border. In 2012, we added 6 additional pumpkin sites for a total of 18. Each site was assigned to one of three habitat management treatments: GRASS CONTROL (pumpkin plot adjacent to a 6 × 60 m grass area, mowed approximately once per month) (2) ALYSSUM (pumpkin plot planted between two 60 m rows of the non-native annual, L. maritima), and (3) PERENNIAL (pumpkin plot planted adjacent to a 6 × 60 m buffer of native perennial wildflowers). These sites were located on 15 farms. Each farm had one pumpkin site except for farms 1, 2, 9, and 10 where both one ALYSSUM and one PERENNIAL treatment site were established. The distances between these plots ranged from 51 m at site 10, to 570 m at site 9.

Table 1 Location of farms studied in 2011 and 2012.

In 2011 we did not evaluate habitat management and each pumpkin site was adjacent to a grassy field boarder (n = 12). In 2012, each pumpkin site was assigned to one of three habitat management treatments: (1) GRASS CONTROL: four rows of pumpkin planted adjacent to a 6 × 60 m grass area, mowed approximately once per month; (2) ALYSSUM: four rows of pumpkin planted between two 60 m rows of L. maritima; and (3) PERENNIAL: pumpkin plots planted adjacent to a 6 × 60 m buffer of native perennials habitat management treatments. Farms 1, 2, 9, and 10 hosted two pumpkin sites in 2012.

	2011	2012	
Farm	Latitude	Longitude	Habitat treatment	Latitude	Longitude	
1	40°54′37.94″N	82°6′35.06″W	Alyssum	40°54′9.69″N	82°6′44.11″W	
1	–	–	Perennial	40°54′38.1″N	82°6′35.5″W	
2	40°55′6.92″N	82°2′57.66″W	Perennial	40°54′58.95″N	82°2′48.14″W	
2	–	–	Alyssum	40°55′5.27″N	82°2′38.15″W	
3	40°56′25.06″N	82°6′58.21″W	–	–	–	
4	41°5′2.65″N	81°57′1.51″W	Control	41°5′3.28″N	81°57′8.13″W	
5	40°42′37.87″N	81°58′16.31″W	Control	40°42′23.5″N	81°57′56.45″W	
6	40°55′17.93″N	81°18′33.26″W	Control	40°55′17.29″N	81°18′31.78″W	
8	–	–	Alyssum	40°58′13.68″N	81°44′25.37″W	
7	–	–	Perennial	40°44′12.27″N	82°11′48.86″W	
9	39°26′5.63″N	83°59′26.59″W	Perennial	39°26′4.39″N	83°59′1.35″W	
9	–	–	Alyssum	39°26′4.01″N	83°59′25.23″W	
10	39°2′50.88″N	82°59′37.4″W	Perennial	39°2′49.35″N	82°59′38.15″W	
10	–	–	Alyssum	39°2′50.88″N	82°59′37.4″W	
11	39°13′13.41″N	83°25′36.81″W	Perennial	39°13′13.41″N	83°25′36.81″W	
12	39°10′58.65″N	83°21′3.09″W	Control	39°10′55″N	83°21′11.37″W	
13	38°59′29.9″N	82°46′4.54″W	Control	38°59′37.44″N	82°45′51.76″W	
14	39°8′16.65″N	82°58′58.47″W	Control	39°8′11.46″N	82°58′59.39″W	
15	–	–	Alyssum	39°24′41.94″N	83°9′27.33″W	

In both years, data was collected from four 60 m rows of jack-o-lantern pumpkins (var. Gladiator), which were established between 10 June and 8 July. No insecticides were applied to the pumpkin plants throughout the study. We refer to each planting as a site. Each site was divided into four 15 m plots that each contained four rows of pumpkin, and all data were collected within these plots. In 2011, one pumpkin site was located per farm (n = 12). In 2012, a total of 18 pumpkin sites were established. Each farm had one site except for farms 1, 2, 9, and 10 where two sites were established (Fig. 1). This was the result of difficulty finding growers willing to host habitat management plantings. Farms 1, 2, 9, and 10 included both an ANNUAL and PERENNIAL treatment pumpkin site (see ‘Habitat management’). The distances between these sites ranged from 51 m at farm 10, to 570 m at farm 9.

Habitat management

In 2012, pumpkin plots within the northern and southern regions were randomly assigned to one of three treatments: (1) GRASS CONTROL: four rows of pumpkin planted adjacent to a 6 × 60 m grass area, mowed approximately once per month; (2) ALYSSUM: four rows of pumpkin planted between two 60 m rows of L. maritima; and (3) PERENNIAL: pumpkin plots planted adjacent to a 6 × 60 m buffer of native perennials (Table 2).

Table 2 Native perennial floral insectaries consisting of 23 forbs and 2 grasses* were established in 6 × 60 m plots in 2010.

The impact of these habitats on pollinator visitation frequency and pollination services was assessed in 2012. The seed mix was designed following Fiedler et al. (2007) and Tuell et al. (2008) to support the production of floral resources throughout the growing season.

	

Establishing non-native annual floral insectaries

In 2012, we planted two rows of L. maritima adjacent to pumpkins at six sites in northern and southern Ohio. For this treatment, one row of L. maritima was established on either side of each four-row pumpkin planting. The L. maritima was started from seed in 72-cell plug trays in a greenhouse in early May and fertilized twice per week for two weeks. The plants were hardened off outside for an additional two weeks before being transplanted with a pottiputki planter (Stand ‘n Plant, Saltsburg, Pennsylvania, USA) between 7–14 June 2012. Plants were watered and Preen Garden Weed Preventer (Lebanon Seaboard Corp., Lebanon, Pennsylvania, USA) was applied. The transplants were watered via drip irrigation and hand containers (∼190 L) twice per week in the field through July.

Establishing native perennial floral insectaries

The perennial insectary was established in fall of 2010 to allow the plants time to establish prior to their evaluation in summer 2012. In October 2010, six farms were selected to establish a 6 × 60 m perennial floral insectary treatment of 23 native forbs and two grasses (Table 2). Each grower cleared the area with field cultivators and herbicide, and rolled the soil flat. We mixed the perennial seeds with sawdust at a ratio of 1:2 and spread 1.3 kg of that mixture at each site to overwinter (Landis, Wratten & Gurr, 2000; Fiedler & Landis, 2007a; Fiedler & Landis, 2007b). The perennial floral insectaries were mowed by the growers once per month to enhance establishment during the 2011 growing season.

Quantifying pollinator assemblages and activity using video surveillance

A modified 4-channel security camera system (Q-see, model no. QSC26404, Anaheim, CA) was used to monitor pollinator activity within two female pumpkin flowers and two male pumpkin flowers within each pumpkin plot (total of 8 female and 8 male flowers observed per site) (Grieshop et al., 2012). Cameras recorded pollinator activity between 0600 h and 1200 h, at 16 frames per second with a playback pixel resolution of 352 × 240 (aspect ratio ∼1.222:1).

In both 2011–12, video surveillance was conducted once during peak bloom in late-July through August. We then omitted pumpkin sites 5, 10, and 13 in 2011 (n = 9 pumpkin plots sampled) due to a wet spring that resulted in an uncommonly late planting, and peak bloom period in September, which we felt was too late to accurately represent the pollinator community that focuses on the pumpkin flower resource pulse. In 2012, pumpkin sites 3p and 7p (n = 16 pumpkin sites sampled) could not be sampled due to heavy weed pressures that drastically reduced pumpkin bloom availability.

After collection, the video footage was transferred to portable hard-drives and stored until viewed on a computer. When a pollinator was observed crossing the plane made by the open corolla, the time of arrival and departure was recorded as a measure of the amount of time spent inside the flower. All pollinators were identified to the lowest taxonomic level possible given the resolution of the video.

Measuring pollen deposition

In 2011 and 2012, we quantified the pollination service provided to each pumpkin site using pollen counts. In 2011 we examined cumulative pollen deposition across three lengths of the pollination window: 2 h (0600–0800 h), 4 h (0600–1000 h) and what we considered the full pollination window of 6 h (0600–1200 h). In 2012 we modified how we measured pollen deposition, collecting data for three individual subsets throughout the pollination window (0600–0800 h, 0800–1000 h, and 1000–1200 h), as well as across the entire period 0600–1200 h. One day prior to the collection of data, mature female flower buds that were at least 5 cm in length and turning deep yellow were located within each site, fitted with a mesh paint strainer bag (Reaves and Co. Durham, North Carolina, USA) as a pollinator excluder, and marked with a step-in poly post (Gempler’s, Madison, Wisconsin, USA). Three (2011) or six (2012) flowers were randomly assigned to each pollen deposition time treatment per pumpkin plot. Bags were left on flowers until the beginning of the treatment time upon which they were removed and pollinators were allowed to access flowers. If the number of flowers needed could not be found for each treatment on the morning of the experiment, we returned within seven days of the first attempt, and in comparable weather conditions to collect additional replicates.

Pollen collection

We designed a simple and inexpensive procedure to collect pollen from stigmas in the field directly after cutting the flower from the pumpkin plant, based in part on the shake and rinse approach of Stanghellini, Schultheis & Ambrose (2002). We used an Aeropress espresso maker and the stock filter discs marked with a 1 × 1 cm grid (Aerobie, Inc., Palo Alto, California, USA) to sieve pollen grains from each collected stigma. Stigmas were placed individually in a 120 mL urine specimen cup with ∼44 mL of a dish soap and water solution (4 drops of dish soap per 2 L of water) and shook vigorously for 20 s. The solution was decanted into a separate cup and the stigma was washed a second time with 70% ethanol. The pollen solution was then poured into the Aeropress, and expunged. The inside of the Aeropress was washed with ethanol so that any pollen that was sticking to the sides was collected on the filter. The filter disc containing the pollen was allowed to dry, packaged individually in labeled petri dishes, and frozen until they were counted under a microscope. Pollen grains from six randomly selected full grid squares, and six partial grid squares were counted and the total pollen load on each filter disc was extrapolated.

Quantifying landscape composition

We obtained aerial image mosaics of each county that contained a research site from the year 2010 (OGRIP, 2010) and uploaded them into ArcMap (version 9.3; ESRI, 2011) and QGIS (version 1.8.0; Quantum GIS Development Team, 2012) to digitize all land cover elements. We determined the area of each distinct landscape feature within 500, 1,000, and 1,500 m radius buffers around the geographic center of each site and ground verified them with a classification system including 22 habitat types. The 22 fine-grain cover types were combined into 7 coarse-grain habitat categories, and the percentages of each habitat type were aggregated as predictor variables within each landscape buffer for analysis (FIELD = percentage of annual field crops; GRASSLAND = percentage of perennial grassland, fallow fields, and pastureland; FORAGE = perennial alfalfa and oats; FRUITVEG = fruit and vegetable cropland; FOREST = woodlands and hedgerows; URBAN = impervious surfaces and buildings; TURF = mowed turfgrass). Total semi-natural habitat (FOREST and GRASSLAND) in each landscape ranged from 10.7–57.7% within a 1,500 m buffer.

Statistical analyses

Visitation frequency

The frequency of total flower visits (fixed factor = bee species), male and female flower visitation (fixed factors = bee species and flower sex), and visitation length (fixed factors = bee species and flower sex) were examined for the three most abundant taxa visiting pumpkin flowers (A. mellifera, Bombus spp., and P. pruinosa) using generalized linear mixed models (glmmadmb function in the glmmADMB package version 0.7.4 in R version 3.0.0) with a Laplace maximum likelihood approximation that allowed for specification of a logistic link function, a negative binomial error distribution for visit frequency data, and a gamma distribution for visit duration data (R Development Core Team, 2013). We also examined how bee visitation to flowers varied by hour of the pollination window using a generalized linear mixed model with the fixed factors flower sex (male or female), bee species (A. mellifera, Bombus spp., or P. pruinosa) and time period (0600–0700, 0701–0800, 0801–0900, 0901–1000, 1001–1100, and 1101–1200 h) and site as a random factor. We used multiple comparisons procedures to contrast the fixed covariates within each model.

Pollen deposition

We modeled the number of pollen grains collected from stigmas of female flowers from three (2011: 2 h (0600–0800 h), 4 h (0600–1000 h) and 6 h (0600–1200 h) or four (2012, 0600–0800 h, 0800–1000 h, and 1000–1200 and 0600–1200 h) time periods using the glmmadmb mixed model function with a negative binomial distribution, and the general linear hypothesis test (glht) function from the multcomp package in R to test for significant differences between time periods.

Habitat management and landscape

To assess whether local habitat management or surrounding landscape composition influenced bee visitation frequency or pollination services we used partial least squares regression analysis (PLS). PLS allows for analysis of models with: (1) multiple response variables, (2) a large number of predictors which may be collinear, and (3) small samples sizes relative the number of possible predictor variables (Carrascal, Galvan & Gordo, 2009). As our landscape variables were proportions of buffer circles, many categories were highly correlated (Appendix S1). PLS reduces sets of predictor and response variables into a smaller set of latent factors.

For 2011, we examined the influence of 21 landscape variables (FOREST, GRASSLAND, FORAGE, FIELD, FRUITVEG, URBAN, AND TURF at 500 m, 1,000 m and 1,500 m radii surrounding each sampling site) on bee visitation frequency and pollen deposition (600–1200 h). In 2012, the influence of the three habitat management treatments (GRASS CONTROL, ALYSSUM and PERENNIAL) was also included in PLS models that examined bee visitation frequency (flower visitation by A. mellifera, Bombus spp., and P. pruinosa), flower visitation by Bombus spp. only (examined as Bombus spp. represented 76.2% of flower visits in 2012), and pollen deposition (600–1200 h). In both years, visitation and pollen deposition were considered separately due to different timing of the experiments.

All predictor variables were centered to a mean of zero and scaled to a standard deviation of one, to give all variables equal weight. The number of factors to be extracted was determined by cross validation using a minimum predicted residual sum of squares (PRESS) as the stop condition. Explanatory variables with a Variable Importance in Projection (VIP) score of >0.8 for a given component were considered significant predictors for that component (SAS Institute Inc, 2011). For each analysis we interpreted up to the first two components (t1 and t2) and only those with a positive Q2 score. Correlation loading plots were used to explore the relationship between the predictor and response variables. These analyses were conducted using the PLS Module of XLSTAT (Addinsoft, Paris, France).

Results

A total of 1,427 A. mellifera (47.7%), 606 Bombus spp. (20.2%), 898 P. pruinosa (30.0%), and 61 other pollinators (2%) were observed in male and female pumpkin flowers in 2011. In 2012 we observed 826 A. mellifera (10.5%), 6,023 Bombus spp. (76.2%), 964 P. pruinosa (12.2%), and 87 other pollinators (1.0%) visiting to pumpkin flowers. Taxa in the other pollinators category included Melissodes bimaculata, Halictidae, Andrenidae and Syrphidae. The number of flower visits by A. mellifera was significantly greater in 2011 than Bombus spp. (z = − 6.23, P < 0.001) or P. pruinosa (z = − 5.85, P < 0.001). The frequency of Bombus spp. and P. pruinosa visitation did not differ (z = 0.65, P = 0.783). In 2012, Bombus spp. were the most frequent visitor, compared to A. mellifera (z = 7.42, P < 0.001) and P. pruinosa (z = − 6.31, P < 0.001). There was no difference in the number of visits by A. mellifera and P. pruinosa (z = − 1.3, P = 0.382).

Male versus female flower visitation

In 2011, A. mellifera visited female flowers more frequently than male flowers (z = − 3.26, P = 0.001), and spent more time in female flowers (z = − 8.1, P < 0.001) (Fig. 2). For Bombus spp., the number of visits to male and female flowers (z = − 0.74, P = 0.461) and the duration of each foraging bout (z = − 0.53, P = 0.594) did not differ (Fig. 2). Similarly, P. pruinosa visit frequency (z = 1.05, P = 0.295) and duration (z = 0.02, P = 0.983) did not vary among male and female flowers (Fig. 2).

Figure 2 The average number of visits (A, B), and average visit duration in minutes of bees (C, D) to male and female flowers in 2011 (A, C) and 2012 (B, D), as observed by video cameras.

Capital letters indicate significant differences within species across flower sex, while lower case letters indicate significant differences among species within a flower sex.

In 2012, A. mellifera again visited female flowers more frequently (z = − 5.42, P < 0.001), and spent more time within them than male flowers (z = − 4.28, P < 0.001) (Fig. 2). Bombus spp. visitation frequency did not vary by flower sex, but bumble bees spent significantly more time in female flowers (z = − 3.24, P = 0.001) (Fig. 2). Squash bees visited male flowers more frequently (z = 2.48, P = 0.013), but individuals spent equal time in both flower sexes (Fig. 2).

Variation among pollinators in male and female flower visitation

In 2011, A. mellifera visited male flowers more frequently than Bombus spp. (z = − 3.1, P = 0.005), but not P. pruinosa (z = − 2.04, P = 0.098). Apis mellifera also visited female flowers more frequently than Bombus spp. (z = − 4.7, P < 0.001) as well as P. pruinosa (z = − 5.06, P < 0.001) (Fig. 2). There was no difference in the number of times Bombus spp. and P. pruinosa visited male flowers (z = 0.91, P = 0.625) or female flowers (z = 0.03, P = 0.999) (Fig. 2). The duration of visits to male flowers did not differ between any bee species, but A. mellifera spent more time in female flowers than both Bombus spp. (z = − 5.28, P < 0.001), and P. pruinosa (z = − 4.56, P < 0.001) (Fig. 2).

In 2012, Bombus spp. visited male and female flowers more often than A. mellifera (z = 8.65, P < 0.001 in male and z = 4.48, P < 0.001 in female flowers), and P. pruinosa (z = − 4.14, P < 0.001 in male and z = − 5.66, P < 0.001 in female flowers) (Fig. 2). Apis mellifera visited female flowers more often (z = − 3.16, P = 0.004) and male flowers less often (z = 2.59, P = 0.024) than P. pruinosa. Honey bees also spent more time in female flowers per visit than Bombus spp. (z = 3.11, P = 0.005) or P. pruinosa (z = − 4.44, P < 0.001) (Fig. 2).

Pollinator activity in flowers throughout the pollination window

In 2011, A. mellifera visitation to female flowers was relatively consistent across the pollination window, with a peak between 0901–1000 h wherein bees visited flowers more frequently than between 0600–0700 h (z = 3.07, P = 0.024) (Figs. 3A and 3C). The time A. mellifera spent inside flowers did not significantly differ by hour. In 2012, A. mellifera flower visitation frequency and duration did not vary by hour (Figs. 3B and 3D).

Figure 3 The average number of visits (A, B), and average visit duration in minutes by bees (C, D) between 0600 h and 1200 h in 2011 (A, C) and 2012 (B, D), as observed by video cameras.

Letters indicate significant differences within species across hour.

In 2011, Bombus spp. visitation frequency was significantly greater after 0700 h (0701–0.800 h: z = 3.44, P = 0.007, 0801–0900 h: z = 3.99, P < 0.001, 0901–1000 h: z = 4.30, P < 0.001, 1001–1100 h: z = 3.69, P = 0.003, 1101–1200 h: z = 3.67, P = 0.003 when compared to 0600–0700 h), yet individuals spent significantly fewer minutes inside flowers after 0700 h (0701–0800 h: z = − 3.42, P = 0.007, 0901–1000 h: z = − 4.64, P < 0.001, 1001–1100 h: z = − 3.73, P = 0.003, 1101–1200 h: z = − 4.83, P < 0.001 when compared to 0600–0700 h) (Figs. 3A and 3C). In 2012, Bombus spp. again visited flowers more frequently after 0700 h (0701–0800 h: z = 6.53, P < 0.001, 0801–0900 h: z = 7.09, P < 0.001, 0901–1000 h: z = 7.29, P < 0.001, 1001–1100 h: z = 6.42, P < 0.001, 1101–1200 h: z = 4.72, P < 0.001 when compared to 0600–0700 h). In 2012, Bombus spp. spent equal time in the flowers throughout the observation period (Figs. 3B and 3D).

In 2011, P. pruinosa visitation frequency did not vary by time of day, but individuals spent significantly more time in the flowers between 0600–0700 h than at times between 0801–1000 h (0801–900 h: z = − 3.84, P = 0.002, 0901–1000 h: z = − 4.84, P < 0.001, 1001–1100 h: z = − 4.22, P < 0.001 when compared to 0600–0700 h) (Figs. 3A and 3C). In 2012, P. pruinosa visitation frequency did not vary by time of day, but individuals spent significantly more minutes inside flowers before 0801 h (z = − 3.53, P < 0.01), and after 1101 h (z = 3.21, P < 0.01 when 1101–1200 h was compared to 0801–0900 h, and z = 3.31, P = 0.012 when 1101–1200 h was compared to 1001–1100 h) (Figs. 3B and 3D).

Pollen deposition throughout the pollination window

In 2011, we found no difference in pollen deposition between 0600–0800 h and 0600–1000 h (z = − 1.18, P = 0.45), between 0600–0800 h and 0600–1200 h (z = 0.26, P = 0.98) or between 0600–1000 h and 0600–1200 h (z = − 1.49, P = 0.29) (Fig. 4). In 2012, the amount of pollen deposited decreased over time, with significantly more pollen grains deposited in flowers between 0600–0800 h than between 1000–1200 h (z = − 3.98, P < 0.001). Total pollen transferred after a full morning (0600–1200 h) was not significantly different from the 0600–0800 h pollination period (z = 0.74, P = 0.871 (Fig. 4).

Figure 4 The average number of pollen grains deposited on pumpkin stigmas across all sites in 2011 was measured at three increasing time intervals: 0600–0800 h, 0600–1000 h, and 0600–1200 h.

The average number of pollen grains across all sites in 2012 was measured at two-hour intervals and across the whole pollination window: 0600–0800 h, 0800–1000 h, 1000–1200 h and 0600–1200 h. Letters indicate significant differences among time periods within an observation year.

Habitat management and landscape influences on visitation frequency and pollen deposition

We found that bee visitation frequency was significantly related to landscape composition variables in 2011 and 2012. In 2011, both t1 and t2 had positive Q2 values. The t1 axis explained an average of 31% and t2 an additional 10% of the variation in visitation by A. mellifera, Bombus spp., and P. pruinosa (Table 3). For t1, 11 variables had a VIP score >0.8 (FOREST 500, 1000 and 1500; FIELD 1000, 1500; FRUITVEG 500, URBAN 1000, 1500 and TURF 500, 1000, 1500). Bombus spp. and P. pruinosa visitation were most strongly correlated with t1 (individual R2 = 0.34 and R2 = 0.39, respectively), and they visited pumpkin flowers in fields surrounded by urbanized areas and forest habitat more frequently than fields surrounded by a significant amount of corn, soybean and fruit and vegetable production (Fig. 5A). For t2, 15 variables had a VIP score >0.8 (FOREST 500, 1000, 1500; GRASSLAND 500, 1000, 1500; FIELD 1000, 1500; FRUITVEG 500; URBAN 500, 1000, 1500; AND TURF 500, 1000, 1500). Apis mellifera was most strongly correlated with t2 (individual R2 = 27.2%). We found that the number of honey bee visits to pumpkin flowers was greater in landscapes with significant amounts of grassland habitat and localized (500 km) urban habitat, and reduced in agricultural landscapes (Fig. 5A).

Figure 5 Correlation maps for the PLS regression of (A) pollinator visitation frequency and landscape variables in 2011 and (B) Bombus spp. visitation frequency and habitat management and landscape variables in 2012.

Only landscape variables with a VIP score of >0.8 for a PLS component (t1 and t2) with a positive Q2 value are shown. In (B) habitat management variables are shown, but only Grass Control had a VIP score of >0.8, indicating that the addition of habitat management did not significantly influence Bombus spp. visitation to pumpkin flowers. Variable abbreviations as follows: Grassland (G), Forest (FO), Field Crops (F), Fruit and Vegetable Crops (FV), Urban (U) and Turf (T).

Table 3 Results of PLS regression analyses examining the influence of landscape variables (2011) and habitat management and landscape variables (2012) on pollinator visitation and pollen deposition.

2011 models included 21 landscape variables (FOREST, GRASSLAND, FORAGE, FIELD, FRUITVEG, URBAN AND TURF at 500 m, 1,000 m and 1,500 m radii surrounding each sampling site); 2012 models included the same landscape variables along with the categorical variable habitat management (GRASS CONTROL, ANNUAL OR PERENNIAL). For each model we report the Q2 (the proportion of the variance in the response variables that can be predicted by the model), the R2Y (the proportion of the variance in the response variable that is explained by the model) and R2X (the proportion of the variance in the matrix of predictor variables that is used in the model) for the first two model components (t1 and t2).

			t 1	t 2	
Year	PLS model	X variable(s)	Q 2	R 2 Y	R 2 X	Q 2	R 2 Y	R 2 X	
2011	Visitation	Apis mellifera	0.17	0.31	0.41	0.12	0.39	0.56	
		Bombus spp.							
		Peponapis pruinosa							
2012	Visitation	Apis mellifera	−0.12	0.12	0.32	−0.28	0.29	0.45	
		Bombus spp.							
		Peponapis pruinosa							
2012	Bumblebee visitation	Bombus spp.	0.08	0.42	0.32	−0.58	0.63	0.42	
2011	Deposition	Pollen grains	0.04	0.47	0.23	−0.44	0.56	0.55	
2012	Deposition	Pollen grains	0.06	0.39	0.31	−0.54	0.47	0.5	

In 2012, we found that neither habitat management nor landscape composition were significant predictors of bee visitation frequency when all three taxa were considered within a PLS model. Given the dominance of Bombus spp. in 2012 (76.2% of flower visits) a second model was examined considering only this group. For Bombus spp. alone we found that t1 explained 41.9% of the variation in bumble bee visitation to pumpkin flowers. A total of 11 variables had VIP scores of >0.8 on the t1 axis (FOREST 500, 1000, 1500; GRASSLAND 1000, 1500; FIELD 500, 1000, 1500, TURF 1000, 1500 and the habitat management variable GRASS CONTROL). We found that bumble bee visitation was highest in pumpkin fields lacking habitat management addition, embedded in landscapes dominated by semi-natural habitat and managed turf, and reduced in agricultural landscapes (Fig. 5B). For Bombus spp. alone, t2 had a negative Q2 value and was not evaluated.

In both 2011 and 2012 pollen deposition was significantly related to landscape composition. In 2011, t1 explained 47% of the variation in pollen deposition; the Q2 value for t2 was negative and thus not examined. Eight variables had a VIP score of >0.8 along t1 (GRASSLAND 500, 1000, 1500; FIELD 500, 1000, 1500; FRUITVEG 1000 AND TURF 1000). Pollen deposition within pumpkin flowers was greater in fields surrounded by significant amounts of grassland habitat and mown turf and reduced in fields embedded in agriculturally-dominated landscapes (Fig. 6A). In 2012, we found that t1 explained 39% of the variation in pollen deposition. Again t2 had a negative Q2 value and was not evaluated. Thirteen variables had a VIP score of >0.8 along the t1 axis (GRASSLAND 500, 1000, 1500; FOREST 500, 1000, 1500; Forage 500, 1000; FIELD 500, 1000, 1500, and URBAN 500, 1000). Similarly to 2011, pollen deposition in pumpkin fields was greater within landscapes dominated by semi-natural and urban habitats and reduced in agriculturally-dominated landscapes (Fig. 6B). As with visitation frequency, we found that the addition of annual or perennial habitat management did not significantly influence pollen deposition.

Figure 6 Correlation maps for the PLS regression of (A) pollen deposition and landscape variables in 2011 and (B) pollen deposition and habitat management and landscape variables in 2012.

Only landscape variables with a VIP >0.8 for the PLS component t1 are shown. In (B) habitat management variables are shown, but none had a VIP score of >0.8, indicating that the addition of habitat management did not significantly influence pollen deposition. Variable abbreviations as follows: Grassland (G), Forest (FO), Forage (FR), Field Crops (F), Fruit and Vegetable Crops (FV), Urban (U) and Turf (T).

Discussion

Large seed set, successful maturation, and fruit weight are highly correlated with the number of pollinator visits to cucurbit flowers (Stanghellini, Ambrose & Schultheis, 1998; Garibaldi et al., 2013) and the amount of pollen transferred to female flowers per visit (Canto-Aguilar & Parra-Tabla, 2000; Winfree et al., 2007; Graças Vidal et al., 2010; Artz & Nault, 2011). Because of this close relationship, research on pollinators of cucurbits has often focused on the abundance of pollinators found inside flowers and the duration of their visitation (Tepedino, 1981; Cane, Minckley & Kervin, 2000; Shuler, Roulston & Farris, 2005; Julier & Roulston, 2009; Nicodemo, Nogueira Couto & De Jong, 2009; Barber, Adler & Bernardo, 2011; Artz, Hsu & Nault, 2011). Our work builds upon these studies using video surveillance to observe pollinator activity throughout the entire 6 h pumpkin pollination window, allowing for documentation of the composition of pollinator fauna visiting male and female flowers as well as their visitation frequency and duration. Further, we were able to measure how the local addition of habitat management as well as larger-scale landscape composition might influence the relationship between pollinator visitation and pollination service within this cropping system.

Visitation frequency of pumpkin pollinators

We found significant variation in the dominant cucurbit pollinator across the two years of our investigation with A. mellifera representing 47.7% of pollinator visits in 2011-significantly more than either Bombus spp. or P. pruinosa. In 2012, Bombus spp. represented 76.2% of pollinator visits to flowers, far more than either A. mellifera or P. pruinosa, which had equivalent visitation frequencies. We saw a nearly a nine-fold increase in the number of bumble bee visits from 573 in 2011 to 5,069 in 2012. Although we do not know what factors contributed to this increase it is only partially explained by a greater number of sites sampled, from nine in 2011 to 16 in 2012. The winter of 2011–12 was among the warmest on record for Ohio, this combined with spring temperatures well above average may have increased survivorship of overwintering queens resulting in a greater number of foraging workers visiting pumpkin fields in 2012.

We aimed to determine if the three bee taxa contributing the majority of pumpkin pollination exhibited variation in how they partitioned their foraging activity among male and female flowers or their temporal use of these resources. Temporally, we found a high level of functional redundancy among this community of pollinators. We did not see much variation in the timing of flower visitation among species, with variation mainly found between the 0600–0700 h when bee activity tended to be lower than the remainder of the pollination window for all taxa. We expected squash bee to be active earlier in the pollination window than other bee species, based on Hurd, Linsley & Michelbacher (1974) who found P. pruinosa to be active 22–55 min before sunrise, and Tepedino (1981) who documented that most pollination provided by squash bee occurred before honey bees became prominent in the crop after 0800 h. However, we found the visitation frequency across the pollination window by this specialist to be relatively consistent with the other taxa. Later activity within flowers could be attributed to P. pruinosa males seeking flowers to shelter in for the afternoon and evening (Michelbacher, Smith & Hurd, 1964; Hurd, Linsley & Michelbacher, 1974).

We did find some differentiation in male versus female flower visitation among bee species. Honey bees visited female flowers more frequently than male flowers and spent more time in female flowers per visit. Bumble bees visited male and female flowers equally in both years, but like A. mellifera spent more time in female flowers in 2012. Similar to our results, Artz & Nault (2011) found that in New York pumpkin fields A. mellifera was more likely to visit female flowers and to spend more time in them. Female flowers produce significantly more nectar than male flowers; collecting nectar is likely to drive this foraging preference (Heinrich, 2004; Seeley, 2009). Unlike the social bees, squash bees visited male flowers more frequently in 2012. Tepedino (1981) also found that P. pruinosa visited more male flowers than female flowers. Similar to other soil and cavity-nesting solitary bees, female P. pruinosa rely more on pollen resources than nectar for solitary brood production.

Pollination services in pumpkin fields

In both 2011 and 2012 we found that the majority of pollen deposition occurred within the first two hours (0600–0800 h) of observation. In fact, there was no difference between the pollen deposited during this two-hour period and the remainder of the pollination window (0600–1200 h) in either year. Graças Vidal et al. (2010) cite 1,500–2,000 pollen grains per flower as a requirement for complete pumpkin pollination. Based on this, the pumpkin plots included in our study received sufficient pollen deposition within just the first two hours of the pollination window.

We saw much higher pollen deposition in 2012 versus 2011, with an average of 4,188 (±294.49 SEM) pollen grains 2012 versus 2,017.59 (±252 SEM) in 2011 deposited between 0600–0800 h. The increase in pollen deposition in 2012 is likely attributable to the far greater visitation frequency by bumble bees. Bumble bees have been reported to be highly efficient pollinators, visiting 4–5 times more flowers per minute than honey bees (Fuchs & Müller, 2004) and carrying up to three times as many pollen grains per visit than A. mellifera or P. pruinosa (Artz, Hsu & Nault, 2011).

Habitat management and pollination services

A key goal of this study was to determine how habitat management influenced the activity of both managed and wild pumpkin pollinators. We found no effect of either annual or perennial habitat management additions on bee visitation frequency or pollen deposition. Pumpkin fields received sufficient pollination services with or without the addition of habitat management.

Although we did not see an increase in ecosystem services delivered by the addition of plant resources, we do not want to convey that habitat management is without value in agricultural landscapes. With regard to our perennial plantings, a time lag may exist between the establishment of the habitat and any change in derived ecosystem services. Our perennial plantings were established in the fall of 2010, and sampled in their second growing season. It is very possible that their impact of on pollination and biocontrol services could change in subsequent years. Further, even if enhanced pest control and pollination are not achieved, perennial plantings have additional environmental benefits. They have been demonstrated to be important for conserving a diverse community of pollinators including those that tend to be most threatened by habitat loss and degradation (Haaland, Naisbit & Bersier, 2011; Wratten et al., 2012; Morandin & Kremen, 2013; Nicholls & Altieri, 2013; Balzan, Bocci & Moonen, 2014; Sardinas & Kremen, 2015; Wood et al., 2015). For example, Kremen & M’Gonigle (2015) found that habitat restoration within hedgerows enhanced the occurrences of native bee and syrphid fly, including taxa with more specialized nesting and foraging requirements and smaller pollinators with reduced mobility among patches.

Furthermore, recent evidence supports that in some agroecosystems these plantings can enhance pollination services. For example, Pereira et al. (2015) examined the utility of intercropping bell pepper with basil on pollination services and found that it increased the richness and abundance of bees visiting pepper flowers. Fruit produced in intercropped plots was also larger and contained more seeds than fruits produced on plots lacking basil plants (Pereira et al., 2015). Blaauw & Isaacs (2014) found that highbush blueberry growing adjacent perennial wildflower habitats exhibited enhanced fruit set, berry weight and mature seeds per berry. Honey bee visitation to blueberry flowers did not increase with wildflower habitat but wild bees and syrphid flies did (Blaauw & Isaacs, 2014). Similarly, Carvalheiro et al. (2012) found that mango orchards near plantings of perennial native plants had greater pollinator visitation and mango fruit production than orchards far from these additions.

Landscape composition and pollinator visitation and pollination services

Landscape variables had a significant influence on bee visitation and pollen deposition. In 2011, bumble bees and squash bees were more abundant in pumpkin fields surrounded by forested and urbanized areas than in fields embedded in agricultural landscapes. In 2012, bumble bees were again more frequent pumpkin flower visitors in fields surrounded by managed turf and semi-natural habitats. We also found that pollen deposition in pumpkin fields was greater within landscapes dominated by semi-natural and urban habitats and reduced in agriculturally-dominated landscapes in 2011–12.

Several studies have found positive relationships between the abundance of semi-natural habitat, landscape heterogeneity, and wild bee abundance and pollination services in crop fields (Steffan-Dewenter et al., 2002; Ricketts et al., 2008; Klein et al., 2012; Kennedy et al., 2013; Andersson et al., 2014; Nayak et al., 2015). For example, Petersen & Nault (2014) used a conditional process modeling approach to illustrate that landscape diversity influenced the impact of bumble bees on pumpkin yield. Bumble bee visits to pumpkin flowers increased yield, but only in highly diverse landscapes (Petersen & Nault, 2014). Xie & An (2014) also found that bumble bee visitation to cucurbit flowers increased with the proportion of surrounding natural habitat, whereas honey bee visitation was unaffected by landscape. We found that landscape did influence honey bee foraging in 2011, with greater visitation by A. mellifera when fields were surrounded by grassland habitat and locally by urban areas. In 2012, however, like Xie & An (2014), we found no effect of landscape composition on honey bee visitation frequency.

In addition, to semi-natural habitat, we found a consistent positive correlation between wild bee visitation, pollen deposition, and the proportion of urban habitat in the surrounding landscape. Managed turf and gardens offer foraging and nesting resources for generalist pollinators like bumble bees (Hagen, Wikelski & Kissling, 2011; Samnegard, Persson & Smith, 2011; Gardiner, Burkman & Prajzner, 2013; Gunnarsson & Federsel, 2014; Parmentier et al., 2014). Additionally, many home and community gardens also produce cucurbit crops and support populations of squash bee. To date, it has not been demonstrated that these habitats serve as a source of either generalist or specialist pollinators to agricultural habitats, but our findings support additional investigation to quantify the value of urban habitats for pollinator conservation and pollination services.

Conclusions

Habitat management seeks to mitigate the negative impacts of agricultural intensification on beneficial arthropods such as predators, parasitoids, and pollinators by providing alternative food and shelter resources (Landis, Wratten & Gurr, 2000; Zehnder et al., 2007). When habitat management practices are incorporated into a farmscape, larger scale landscape composition and heterogeneity structure the pool of beneficial species supplied to the floral insectary, which ultimately influences the arthropod-mediated ecosystem services they are able to support (Isaacs et al., 2009; Batáry et al., 2011; Concepción et al., 2012; Rodriguez-Saona, Blaauw & Isaacs, 2012). Tscharntke et al. (2012) introduced the Intermediate Landscape Complexity Hypothesis, which states that in highly heterogeneous landscapes (>20% non-crop habitats), stable populations of beneficial organisms already exist which limited the effect of local habitat management; and extremely simplified landscapes (<1% non-crop habitats) do not have enough supporting habitats for a substantial species pool to take advantage of local habitat amendments. As such, local habitat management is theoretically most useful to enhance arthropod-mediated ecosystem services within intermediately-complex landscapes. In 2012, when we evaluated the habitat management plantings only three landscapes fell into this intermediate landscape category, with all other sites having >20% non-crop habitat. To advance our understanding of the role of habitat management in provisioning ecosystem services, future work should explore whether we find a landscape threshold at which adding habitat resoruces on-farm alters the activity of insects that provide pollination or biocontrol services. Understanding these relationships would aid in the development of agri-environment schemes to enhance habitat for beneficial arthropods within the US where broad-scale implementataion of such plans lag behind those underway in the UK and continental Europe.

Supplemental Information

Supplemental Information 1 Raw data with keys to the legand

Click here for additional data file.

Supplemental Information 2 Summary file of the 2011 pollen data

Click here for additional data file.

Supplemental Information 3 Summary file of the 2011 pollinator data

Click here for additional data file.

Supplemental Information 4 Summary file of 2012 pollen data

Click here for additional data file.

Supplemental Information 5 Summary file of 2012 pollinator data

Click here for additional data file.

Appendix S1 Appendix 1

Click here for additional data file.

Thank you to all of our grower-collaborators who assisted with the establishment and maintenance of our pumpkin and floral insectary plots as well as Brad Bergefurd, Jim Jasinski and Dr. Celeste Welty for providing help in finding such great collaborators. We thank Andrea Kautz and Ted Green for their positive attitudes and attention to detail as summer research assistants. Guidance with statistical analysis was provided by Dr. Alain Zurr of Highland Statistics Ltd. We also thank Dr. Larry Phelan, Dr. Robin Taylor and Diego Rincon for additional assistance with statistical analysis and graphics programs. Thank you to Chris Riley for proofreading assistance. We thank our Editor Mattias Jonsson, reviewer Manu Saunders, and a second anonymous reviewer for their helpful feedback on an earlier draft of this manuscript.

Additional Information and Declarations

Competing Interests

Author Contributions

The authors declare there are no competing interests.

Benjamin W. Phillips conceived and designed the experiments, performed the experiments, analyzed the data, wrote the paper, prepared figures and/or tables, reviewed drafts of the paper.

Mary M. Gardiner conceived and designed the experiments, analyzed the data, contributed reagents/materials/analysis tools, wrote the paper, prepared figures and/or tables, reviewed drafts of the paper.

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
