# Peer review of "Use of video surveillance to measure the influences of habitat management and landscape composition on pollinator visitation and pollen deposition in pumpkin (Cucurbita pepo) agroecosystems"

_PeerJ, doi:10.7717/peerj.1342_

## Round 0.1 · original submission · Major Revisions

I agree with both reviewers that the article currently does not quite match the title and stated aims. It should have a stronger focus on the effects of habitat management and landscape. Other aspects could be condensed - the paper is currently quite long.

·

Basic reporting

This is an interesting study that aims to assess how local habitat management and landscape composition influence flower visitation on pumpkin flowers. In general, I think it is of interest to the pollination literature. Habitat management schemes based on establishing floral arrays in farms, gardens or urban spaces are increasingly being recommended as a pollinator conservation initiative, but very little research exists on how such schemes influence pollinator visitation, or what long-term effects they may have.
However, I found the manuscript hard to follow and it requires some editing for clarity – comments are detailed below. In particular, the results don’t appear to match your study goals. The title and Intro state you will be assessing how habitat mgmt treatments and landscape composition (separately or together? I wasn’t clear on this) affect bee visitation to pumpkins. However, the majority of your results simply discuss differences in bee visitation frequency between species and years, and there are very few data presented on how the habitat mgmt treatments affected bee visitation. I wonder if you have tried to combine two separate papers here – one on visitation/pollen deposition differences between species, and one on the effects of landscape/habitat mgmt on bee visitation.
Introduction: This is generally good, but is a bit long. You could easily condense some of these sections to provide a clearer focus. Your study goal is to assess habitat management effects on pollinator visitation, so much of the general ‘pollinator decline’ info at the start could be summarised more briefly.

General:
Line 18-19: are some words missing here? The punctuation and sentence structure are odd
Line 19: change to “In 2011, A. mellifera made significantly more floral visits...”
Line 26: change to “In both 2011 and 2012...”
Line 65: reword first line to “several potential drivers of population decline in pollinator species have been...”
Lines 65-71: cite the references after each concept they relate to, rather than as a block at the end of the sentence. For example: “ ...including pesticide use (ref, ref), pathogen and parasite infection (ref), exposure to heavy metals (ref, ref)...etc.”
Line 75: “though” should be “through”
Line 87-90: This sentence needs restructuring for clarity, as the current structure doesn’t match the number of clauses.
Lines 110-114: These two sentences (referring to Garibaldi & Ricketts studies) could be combined to one sentence, as the two studies have similar findings
Line 122: Habitat is misspelled
Line 131: remove “the value of”
Lines 132-133: reword to “because they are visited by managed (A. meliifera) and wild (Bombus spp.) social bees as well as a wild solitary specialist pollinator..”
Lines 136-137: remove “the quantities and quality of” and insert “efficiency” before the citation.
Line 138: “habitat management addition” is not correct usage. You would either say “habitat management” or “addition of floral habitat” or something like that
Line 142: “boarder” should be “border”
Line 264: reword to “Total semi-natural habitat (FOREST and GRASSLAND) in each landscape ranged from 10.7-57.7% within 1500 m.”
Line 479: insert “flowers” after the word female
Line 482-43: you have doubled up on the word “that”
Lines 485:486: this sentence isn’t clear to me. What/who don’t the bees co-exist with?
Line 581: habitat is misspelled

Experimental design

Methods: Good. Suitable information has been included to enable replication. I found it hard to follow all the changes in sites that were added/dropped between years. Perhaps you could add “n” values to the list of sites in Table 1 for each year/treatment combination.
Lines 167-174: I found this paragraph a bit confusing. You say each plot consisted of 4 rows. And later that each pot was divided into 4 sub-plots...so was 1 row a sub-plot?
Lines 171-174: I think you could move these sentences to the next section where you discuss the annual/perennial plots. Also, is there a reason why sites 2, 9 and 10 had two treatments, while other sites were randomly assigned to one treatment?
Table 1: Please also explain briefly in the caption why there are two sites listed for 2, 9 and 10. Or perhaps you could specify in the table which one was Annual and Perennial for the 2012 coordinates.

Validity of the findings

Results: There is a lot of information here about visitation/pollen deposition, but not much about the landscape comp/habitat mgmt effects. The title, abstract and Intro suggest that the study will identify how habitat mgmt & landscape comp influence bee visitation, yet the results here mainly discuss differences in visitation between years and species. Section 3.4 gives the only results that are really relevant to your goal, but they also hardly mention the habitat mgmt side of the study. I think you could condense a lot of the differences between species to focus on your main issue, the effect of landscape/habitat mgmt. I can only find one mention (Line 415) of the habitat mgmt treatments; other than that, there are very few results showing how visitation differed between annual/perennial/control treatments, yet this was apparently the goal of your study.
Line 318: Were any other pollinators seen visiting flowers on camera footage? Did you notice an increase/decrease in other pollinator sp. abundance between habitat mgmt treatments?
Lines 415-418: you direct the reader to Figure 5 here, but I can’t find where the habitat mgmt treatments are represented in this figure, or fig 6 – the variables shown on the graph are all landscape variables. If the habitat mgmt treatments aren’t shown, you should explain why in the captions.
Discussion:
Line 454-459: could the increase in Bombus have been influenced by the additional sites you sampled in 2012, but not 2011?
Lines 509-514: Could the lack of effect be an artefact of time? As you only waited a few months after planting the perennials to sample, would there be a time lag before you saw an increase in bee populations?

Reviewer 2 ·

Basic reporting

Although contains some typos and missing words will require some editing, I enjoyed the reading of this manuscript that is clear and relatively easy to follow. The methods and results are generally well presented and the figures are clear and relevant. The context and the rational of the study are clearly stated in the introduction and concepts and ideas well supported by the literature.
In the introduction, the authors do not present the concept of density dependence and the potential competition for pollination services among floral patches. I believe that this is important to consider, at least discussing, in the case of habitat management and establishment of floral insectaries adjacent to farm fields. The method and the experimental design are adequately described with figure 1 supporting site location and treatment distribution.

Experimental design

All the metrics and the methods used are clearly described. Nevertheless, I was wondering if there was a reference for the pollen extraction solution (L-242-244), if so it should be mentioned. A question I had while reading the manuscript was if the numbers of A. mellifera hives where comparable across years and sites?

Validity of the findings

In the method or in the results, the data related to the landscape are not available. A summary table presenting the correlation matrices between habitat categories and scales (buffers) would help to better understand the structure and composition of the studied landscapes.
Overall, the statistical methods used are sound and seems to be rigorously applied. Nevertheless, in the description of the GLMM, the fixed effects are stated, but the random effects need to be clarified. The glht at L-289 is used to test for statistical difference between time periods, not to "separate means". In your method, can you provide a reference to justify the threshold of 8 for the VIP in the PLS?
The results are clearly presented and in line with the analysis. In such study however, I consider that the relationship between pollinator visit (frequency and duration) and pollen deposition should be tested. This was also partly suggested as an aim of the study L-447-448, but not tested.
The discussion is adequate and links the results with the literature, providing some constructive ideas to interpret the results and highlighting the gaps that need to be filled.

---

## Round 0.2 · accepted · Accept

You have done a very good job responding to the comments by the referees and making appropriate changes to the manuscript.